# Empirical Risk Minimization
# Under Fairness Constraints

**Michele Donini**[1], **Luca Oneto** [2],
**Shai Ben-David**[3], **John Shawe-Taylor**[4], **Massimiliano Pontil**[1,4]
[1]Istituto Italiano di Tecnologia (Italy) [2]University of Genoa (Italy),
[3]University of Waterloo (Canada), [4]University College London (UK)

## Abstract

We address the problem of algorithmic fairness: ensuring that sensitive information does not unfairly influence the outcome of a classifier. We present an approach based on empirical risk minimization, which incorporates a fairness constraint into the learning problem. It encourages the conditional risk of the learned classifier to be approximately constant with respect to the sensitive variable. We derive both risk and fairness bounds that support the statistical consistency of our methodology. We specify our approach to kernel methods and observe that the fairness requirement implies an orthogonality constraint which can be easily added to these methods. We further observe that for linear models the constraint translates into a simple data preprocessing step. Experiments indicate that the method is empirically effective and performs favorably against state-of-the-art approaches.

## 1 Introduction

In recent years there has been a lot of interest on algorithmic fairness in machine learning see, e.g., [1–13] and references therein. The central question is how to enhance supervised learning algorithms with fairness requirements, namely ensuring that sensitive information (e.g. knowledge about the ethnic group of an individual) does not 'unfairly' influence the outcome of a learning algorithm. For example if the learning problem is to decide whether a person should be offered a loan based on her previous credit card scores, we would like to build a model which does not unfairly use additional sensitive information such as race or sex. Several notions of fairness and associated learning methods have been introduced in machine learning in the past few years, including Demographic Parity [14], Equal Odds and Equal Opportunities [2], Disparate Treatment, Impact, and mistreatment [3]. The underlying idea behind such notions is to balance decisions of a classifier among the different sensitive groups and label sets.

In this paper, we build upon the notion Equal Opportunity (EO) which defines fairness as the requirement that the true positive rate of the classifier is the same across the sensitive groups. In Section 2 we introduce a generalization of this notion of fairness which constrains the conditional risk of a classifier, associated to positive labeled samples of a group, to be approximately constant with respect to group membership. The risk is measured according to a prescribed loss function and approximation parameter $\epsilon$. When the loss is the misclassification error and $\epsilon = 0$ we recover the notion EO above. We study the problem of minimizing the expected risk within a prescribed class of functions subject to the fairness constraint. As a natural estimator associated with this problem, we consider a modified version of Empirical Risk Minimization (ERM) which we call Fair ERM (FERM). We derive both risk and fairness bounds, which support that FERM is statistically consistent, in a certain sense which we explain in the paper in Section 2.2. Since the FERM approach is impractical due to the non-convex nature of the constraint, in the same section we propose a surrogate convex FERM problem which relates, under a natural condition, to the original goal of

minimizing the misclassification error subject to a relaxed EO constraint. We further observe that our condition can be empirically verified to judge the quality of the approximation in practice. As a concrete example of the framework, in Section 3 we describe how kernel methods such as support vector machines (SVMs) can be enhanced to satisfy the fairness constraint. We observe that a particular instance of the fairness constraint for $\epsilon = 0$ reduces to an orthogonality constraint. Moreover, in the linear case, the constraint translates into a preprocessing step that implicitly imposes the fairness requirement on the data, making fair any linear model learned with them. We report numerical experiments using both linear and nonlinear kernels, which indicate that our method improves on the state-of-the-art in four out of five datasets and is competitive on the fifth dataset. Additional technical steps and experiments are presented in the supplementary material.

In summary the contributions of this paper are twofold. First, we outline a general framework for empirical risk minimization under fairness constraints. The framework can be used as a starting point to develop specific algorithms for learning under fairness constraints. As a second contribution, we shown how a linear fairness constraint arises naturally in the framework and allows us to develop a novel convex learning method that is supported by consistency properties both in terms of EO and risk of the selected model, performing favorably against state-of-the-art alternatives on a series of benchmark datasets.

**Related Work.** Work on algorithmic fairness can be divided in three families. Methods in the first family modify a pretrained classifier in order to increase its fairness properties while maintaining as much as possible the classification performance: [2, 15–17] are examples of these methods, however no consistency properties nor comparison with state-of-the-art proposals are provided. Methods in the second family enforce fairness directly during the training step, e.g. [3, 18–26]. However they either provide non-convex approaches to the solution of the problem or derive consistency results for the non-convex formulation, resorting later to a convex approach which is not theoretically grounded; The third family of methods implements fairness by modifying the data representation and then employs standard machine learning methods: [4, 7, 27–30] are examples of these methods. Again, no consistency property nor comparison with state-of-the-art proposal are provided. Our method belongs to the second family, in that it directly optimizes a fairness constraint related to the notion of EO discussed above. Furthermore, in the case of linear models, our method translates into an efficient preprocessing of the input data, with methods in the third family. Finally our method can be extended to deal with other frameworks like the multitask learning one [31]. As we shall see, our approach is statistically consistent and performs favorably against the state-of-the-art. We are aware that other convex methods exist, e.g. [32–34] which, however, do not compare with other state-of-the-art solutions and do not provide consistency. In this sense, an exception is [1] but, contrarily to our proposal, do not enforce a fairness constraint directly in the learning phase. We note that a more detailed comparison between our proposal and state-of-the-art is reported in the supplementary material, Section C.

## 2 Fair Empirical Risk Minimization

In this section, we present our approach to learning with fairness. We begin by introducing our notation. We let $\mathcal{D} = \{(\boldsymbol{x}_1, s_1, y_1), \ldots, (\boldsymbol{x}_n, s_n, y_n)\}$ be a sequence of $n$ samples drawn independently from an unknown probability distribution $\mu$ over $\mathcal{X} \times \mathcal{S} \times \mathcal{Y}$, where $\mathcal{Y} = \{-1, +1\}$ is the set of binary output labels, $\mathcal{S} = \{a, b\}$ represents group membership among two groups[1] (e.g. 'female' or 'male'), and $\mathcal{X}$ is the input space. We note that the input $\boldsymbol{x} \in \mathcal{X}$ may further contain or not the sensitive feature $s \in \mathcal{S}$ in it. We also denote by $\mathcal{D}^{+,g} = \{(\boldsymbol{x}_i, s_i, y_i) : y_i{=}1, s_i{=}g\}$ for $g \in \{a, b\}$ and $n^{+,g} = |\mathcal{D}^{+,g}|$. Let us consider a function (or model) $f : \mathcal{X} \to \mathbb{R}$ chosen from a set $\mathcal{F}$ of possible models. The error (risk) of $f$ in approximating $\mu$ is measured by a prescribed loss function $\ell : \mathbb{R} \times \mathcal{Y} \to \mathbb{R}$. The risk of $f$ is defined as $L(f) = \mathbb{E}\left[\ell(f(\boldsymbol{x}), y)\right]$. When necessary we will indicate with a subscript the particular loss function used, i.e. $L_p(f) = \mathbb{E}\left[\ell_p(f(\boldsymbol{x}), y)\right]$.

The purpose of a learning procedure is to find a model that minimizes the risk. Since the probability measure $\mu$ is usually unknown, the risk cannot be computed, however we can compute the empirical risk $\hat{L}(f) = \hat{\mathbb{E}}[\ell(f(\boldsymbol{x}), y)]$, where $\hat{\mathbb{E}}$ denotes the empirical expectation. A natural learning strategy, called Empirical Risk Minimization (ERM), is then to minimize the empirical risk within a prescribed set of functions.

## 2.1 Fairness Definitions

In the literature there are different definitions of fairness of a model or learning algorithm [1-3, 3], but there is not yet a consensus on which definition is most appropriate. In this paper, we introduce a general notion of fairness which encompasses some previously used notions and it allows to introduce new ones by specifying the loss function used below.

**Definition 1.** *Let $L^{+,g}(f) = \mathbb{E}[\ell(f(\boldsymbol{x}), y)|y=1, s=g]$ be the risk of the positive labeled samples in the g-th group, and let $\epsilon \in [0,1]$. We say that a function f is $\epsilon$-fair if $|L^{+,a}(f) - L^{+,b}(f)| \leq \epsilon$.*

This definition says that a model is fair if it commits approximately the same error on the positive class independently of the group membership. That is, the conditional risk $L^{+,g}$ is approximately constant across the two groups. Note that if $\epsilon = 0$ and we use the hard loss function, $\ell_h(f(\boldsymbol{x}), y) = \mathbb{1}_{\{yf(\boldsymbol{x}) \leq 0\}}$, then Definition 1 is equivalent to definition of EO proposed by [2], namely

$$\mathbb{P}\{f(\boldsymbol{x}) > 0 \mid y = 1, s = a\} = \mathbb{P}\{f(\boldsymbol{x}) > 0 \mid y = 1, s = b\}. \tag{1}$$

This equation means that the true positive rate is the same across the two groups. Furthermore, if we use the linear loss function $\ell_l(f(\boldsymbol{x}), y) = (1 - yf(\boldsymbol{x}))/2$ and set $\epsilon = 0$, then Definition 1 gives

$$\mathbb{E}[f(\boldsymbol{x}) \mid y = 1, s = a] = \mathbb{E}[f(\boldsymbol{x}) \mid y = 1, s = b]. \tag{2}$$

By reformulating this expression we obtain a notion of fairness introduced in [1]

$$\sum_{g \in \{a,b\}} \big| \mathbb{E}[f(\boldsymbol{x}) \mid y = 1, s = g] - \mathbb{E}[f(\boldsymbol{x}) \mid y = 1] \big| = 0.$$

Yet another implication of Eq. (2) is that the output of the model is uncorrelated with respect to the group membership conditioned on the label being positive [35], that is, for every $g \in \{a, b\}$, we have

$$\mathbb{E}\big[f(\boldsymbol{x})\mathbb{1}_{\{s=g\}} \mid y = 1\big] = \mathbb{E}\big[f(\boldsymbol{x})|y=1\big]\mathbb{E}\big[\mathbb{1}_{\{s=g\}} \mid y = 1\big].$$

Finally, we observe that our approach naturally generalizes to other fairness measures that are based on conditional probabilities, e.g. equal odds [2] (see the supplementary material, Section A). Specifically, we would require in Definition 1 that $|L^{y,a}(f) - L^{y,b}(f)| \leq \epsilon$ for both $y \in \{-1, 1\}$.

## 2.2 Fair Empirical Risk Minimization

In this paper, we aim at minimizing the risk subject to a fairness constraint. Specifically, we consider the problem

$$\min\Big\{L(f) : f \in \mathcal{F}, \ \big|L^{+,a}(f) - L^{+,b}(f)\big| \leq \epsilon\Big\}, \tag{3}$$

where $\epsilon \in [0,1]$ is the amount of unfairness that we are willing to bear. Since the measure $\mu$ is unknown we replace the deterministic quantities with their empirical counterparts. That is, we replace Problem 3 with

$$\min\Big\{\hat{L}(f) : f \in \mathcal{F}, \ \big|\hat{L}^{+,a}(f) - \hat{L}^{+,b}(f)\big| \leq \hat{\epsilon}\Big\}, \tag{4}$$

where $\hat{\epsilon} \in [0,1]$. We will refer to Problem 4 as FERM.

We denote by $f^*$ a solution of Problem 3, and by $\hat{f}$ a solution of Problem 4. In this section we will show that these solutions are linked one to another. In particular, if the parameter $\hat{\epsilon}$ is chosen appropriately, we will show that, in a certain sense, the estimator $\hat{f}$ is consistent. In order to present our observations, we require that it holds with probability at least $1 - \delta$ that

$$\sup_{f \in \mathcal{F}} \big|L(f) - \hat{L}(f)\big| \leq B(\delta, n, \mathcal{F}) \tag{5}$$

where the bound $B(\delta, n, \mathcal{F})$ goes to zero as $n$ grows to infinity if the class $\mathcal{F}$ is learnable with respect to the loss [see e.g. 36, and references therein]. For example, if $\mathcal{F}$ is a compact subset of linear separators in a Reproducing Kernel Hilbert Space (RKHS), and the loss is Lipschitz in its first argument, then $B(\delta, n, \mathcal{F})$ can be obtained via Rademacher bounds [see e.g. 37]. In this case $B(\delta, n, \mathcal{F})$ goes to zero at least as $\sqrt{1/n}$ as $n$ grows to infinity, where $n = |\mathcal{D}|$.

We are ready to state the first result of this section (proof is reported in supplementary material, Section B).

**Theorem 1.** *Let $\mathcal{F}$ be a learnable set of functions with respect to the loss function $\ell : \mathbb{R} \times \mathcal{Y} \to \mathbb{R}$, let $f^*$ be a solution of Problem (3) and let $\hat{f}$ be a solution of Problem (4) with*

$$\hat{\epsilon} = \epsilon + \sum_{g \in \{a,b\}} B(\delta, n^{+,g}, \mathcal{F}). \tag{6}$$

*With probability at least $1 - 6\delta$ it holds simultaneously that*

$$L(\hat{f}) - L(f^*) \leq 2B(\delta, n, \mathcal{F}) \quad and \quad \left| L^{+,a}(\hat{f}) - L^{+,b}(\hat{f}) \right| \leq \epsilon + 2 \sum_{g \in \{a,b\}} B(\delta, n^{+,g}, \mathcal{F}).$$

A consequence of the first statement of Theorem 1 is that as $n$ tends to infinity $L(\hat{f})$ tends to a value which is not larger than $L(f^*)$, that is, FERM is consistent with respect to the risk of the selected model. The second statement of Theorem 1, instead, implies that as $n$ tends to infinity we have that $\hat{f}$ tends to be $\epsilon$-fair. In other words, FERM is consistent with respect to the fairness of the selected model.

Thanks to Theorem 1 we can state that $f^*$ is close to $\hat{f}$ both in term of its risk and its fairness. Nevertheless, our final goal is to find an $f_h^*$ which solves the following problem

$$\min \left\{ L_h(f) : f \in \mathcal{F}, \; \left| L_h^{+,a}(f) - L_h^{+,b}(f) \right| \leq \epsilon \right\}. \tag{7}$$

Note that the objective function in Problem 7 is the misclassification error of the classifier $f$, whereas the fairness constraint is a relaxation of the EO constraint in Eq. (1). Indeed, the quantity $\left| L_h^{+,a}(f) - L_h^{+,b}(f) \right|$ is equal to

$$\left| \mathbb{P} \left\{ f(\boldsymbol{x}) > 0 \mid y = 1, s = a \right\} - \mathbb{P} \left\{ f(\boldsymbol{x}) > 0 \mid y = 1, s = b \right\} \right|. \tag{8}$$

We refer to this quantity as difference of EO (DEO).

Although Problem 7 cannot be solved, by exploiting Theorem 1 we can safely search for a solution $\hat{f}_h$ of its empirical counterpart

$$\min \left\{ \hat{L}_h(f) : f \in \mathcal{F}, \; \left| \hat{L}_h^{+,a}(f) - \hat{L}_h^{+,b}(f) \right| \leq \hat{\epsilon} \right\}. \tag{9}$$

Unfortunately Problem 9 is a difficult nonconvex nonsmooth problem, and for this reason it is more convenient to solve a convex relaxation. That is, we replace the hard loss in the risk with a convex loss function $\ell_c$ (e.g. the Hinge loss $\ell_c = \max\{0, \ell_l\}$) and the hard loss in the constraint with the linear loss $\ell_l$. In this way, we look for a solution $\hat{f}_c$ of the convex FERM problem

$$\min \left\{ \hat{L}_c(f) : f \in \mathcal{F}, \; \left| \hat{L}_l^{+,a}(f) - \hat{L}_l^{+,b}(f) \right| \leq \hat{\epsilon} \right\}. \tag{10}$$

Note that this approximation of the EO constraint correspond to matching the first order moment. Other works tries to match the second order moment [20] or potentially infinitely many moments [38] but these approaches result in non-convex approaches.

The questions that arise here are whether, $\hat{f}_c$ is to $\hat{f}_h$, how much, and under which assumptions. The following theorem sheds some lights on these issues (proof is reported in supplementary material, Section B).

**Proposition 1.** *If $\ell_c$ is the Hinge loss then $\hat{L}_h(f) \leq \hat{L}_c(f)$. Moreover, if for $f : \mathcal{X} \to \mathbb{R}$ the following condition is true*

$$\tfrac{1}{2} \sum_{g \in \{a,b\}} \left| \hat{\mathbb{E}} \left[ \mathrm{sign}\left( f(\boldsymbol{x}) \right) - f(\boldsymbol{x}) \mid y = 1, s = g \right] \right| \leq \hat{\Delta}, \tag{11}$$

*then it also holds that*

$$\left| \hat{L}_h^{+,a}(f) - \hat{L}_h^{+,b}(f) \right| \leq \left| \hat{L}_l^{+,a}(f) - \hat{L}_l^{+,b}(f) \right| + \hat{\Delta}.$$

The first statement of Proposition 1 tells us that exploiting $\ell_c$ instead of $\ell_h$ is a good approximation if $\hat{L}_c(\hat{f}_c)$ is small. The second statement of Proposition 1, instead, tells us that if the hypothesis of inequality (11) holds, then the linear loss based fairness is close to the EO. Obviously the smaller $\hat{\Delta}$ is, the closer they are. Inequality (11) says that the functions $\mathrm{sign}\left( f(\boldsymbol{x}) \right)$ and $f(\boldsymbol{x})$ distribute,

on average, in a similar way. This condition is quite natural and it has been exploited in previous work [see e.g. 39]. Moreover, in Section 4 we present experiments showing that $\hat{\Delta}$ is small.

The bound in Proposition 1 may be tighten by using different nonlinear approximations of EO [see e.g. 7]. However, the linear approximation proposed in this work gives a convex problem, and as we shall see in Section 5, works well in practice.

In summary, the combination of Theorem 1 and Proposition 1 provides conditions under which a solution $\hat{f}_c$ of Problem 4, which is convex, is close, *both in terms of classification accuracy and fairness*, to a solution $f_h^*$ of Problem 7, which is our final goal.

# 3   Fair Learning with Kernels

In this section, we specify the FERM framework to the case that the underlying space of models is a reproducing kernel Hilbert space (RKHS) [see e.g. 40, 41, and references therein]. We let $\kappa : \mathcal{X} \times \mathcal{X} \to \mathbb{R}$ be a positive definite kernel and let $\phi : \mathcal{X} \to \mathbb{H}$ be an induced feature mapping such that $\kappa(\boldsymbol{x}, \boldsymbol{x}') = \langle \phi(\boldsymbol{x}), \phi(\boldsymbol{x}') \rangle$, for all $\boldsymbol{x}, \boldsymbol{x}' \in \mathcal{X}$, where $\mathbb{H}$ is the Hilbert space of square summable sequences. Functions in the RKHS can be parametrized as

$$f(\boldsymbol{x}) = \langle \boldsymbol{w}, \phi(\boldsymbol{x}) \rangle, \quad \boldsymbol{x} \in \mathcal{X}, \tag{12}$$

for some vector of parameters $\boldsymbol{w} \in \mathbb{H}$. In practice a bias term (threshold) can be added to $f$ but to ease our presentation we do not include it here.

We solve Problem (10) with $\mathcal{F}$ a ball in the RKHS and employ a convex loss function $\ell$. As for the fairness constraint we use the linear loss function, which implies the constraint to be convex. Let $\boldsymbol{u}_g$ be the barycenter in the feature space of the positively labelled points in the group $g \in \{a, b\}$, that is

$$\boldsymbol{u}_g = \tfrac{1}{n^{+,g}} \sum_{i \in \mathcal{I}^{+,g}} \phi(\boldsymbol{x}_i), \tag{13}$$

where $\mathcal{I}^{+,g} = \{i : y_i = 1, s_i = g\}$. Then using Eq. (18) the constraint in Problem (10) takes the form $\big| \langle \boldsymbol{w}, \boldsymbol{u}_a - \boldsymbol{u}_b \rangle \big| \le \epsilon$.

In practice, we solve the Tikhonov regularization problem

$$\min_{\boldsymbol{w} \in \mathbb{H}} \sum_{i=1}^n \ell(\langle \boldsymbol{w}, \phi(\boldsymbol{x}_i) \rangle, y_i) + \lambda \|\boldsymbol{w}\|^2 \quad \text{s.t.} \ \big| \langle \boldsymbol{w}, \boldsymbol{u} \rangle \big| \le \epsilon \tag{14}$$

where $\boldsymbol{u} = \boldsymbol{u}_a - \boldsymbol{u}_b$ and $\lambda$ is a positive parameter which controls model complexity. In particular, if $\epsilon = 0$ the constraint in Problem (14) reduces to an orthogonality constraint that has a simple geometric interpretation. Specifically, the vector $\boldsymbol{w}$ is required to be orthogonal to the vector formed by the difference between the barycenters of the positive labelled input samples in the two groups.

By the representer theorem [42], the solution to Problem (14) is a linear combination of the feature vectors $\phi(\boldsymbol{x}_1), \dots, \phi(\boldsymbol{x}_n)$ and the vector $\boldsymbol{u}$. However, in our case $\boldsymbol{u}$ is itself a linear combination of the feature vectors (in fact only those corresponding to the subset of positive labeled points) hence $\boldsymbol{w}$ is a linear combination of the input points, that is $\boldsymbol{w} = \sum_{i=1}^n \alpha_i \phi(\boldsymbol{x}_i)$. The corresponding function used to make predictions is then given by $f(\boldsymbol{x}) = \sum_{i=1}^n \alpha_i \kappa(\boldsymbol{x}_i, \boldsymbol{x})$. Let $K$ be the Gram matrix. The vector of coefficients $\boldsymbol{\alpha}$ can then be found by solving

$$\min_{\boldsymbol{\alpha} \in \mathbb{R}^n} \left\{ \sum_{i=1}^n \ell \left( \sum_{j=1}^n K_{ij} \alpha_j, y_i \right) + \lambda \sum_{i,j=1}^n \alpha_i \alpha_j K_{ij} \quad \text{s.t.} \ \left| \sum_{i=1}^n \alpha_i \left[ \frac{1}{n^{+,a}} \sum_{j \in \mathcal{I}^{+,a}} K_{ij} - \frac{1}{n^{+,b}} \sum_{j \in \mathcal{I}^{+,b}} K_{ij} \right] \right| \le \epsilon \right\}.$$

In our experiments below we consider this particular case of Problem (14) and furthermore choose the loss function $\ell_c$ to be the Hinge loss. The resulting method is an extension of SVM. The fairness constraint and, in particular, the orthogonality constraint when $\epsilon = 0$, can be easily added within standard SVM solvers[2].

It is instructive to consider Problem (14) when $\phi$ is the identity mapping (i.e. $\kappa$ is the linear kernel on $\mathbb{R}^d$) and $\epsilon = 0$. In this special case we can solve the orthogonality constraint $\langle \boldsymbol{w}, \boldsymbol{u} \rangle = 0$ for $w_i$, where the index $i$ is such that $|u_i| = \|\boldsymbol{u}\|_\infty$, obtaining that $w_i = -\sum_{j=1, j \ne i}^d w_j \frac{u_j}{u_i}$. Consequently the linear model rewrites as $\sum_{j=1}^d w_j x_j = \sum_{j=1, j \ne i}^d w_j (x_j - x_i \frac{u_j}{u_i})$. In this way, we then see the

fairness constraint is implicitly enforced by making the change of representation $\boldsymbol{x} \mapsto \tilde{\boldsymbol{x}} \in \mathbb{R}^{d-1}$, with

$$\tilde{x}_j = x_j - x_i \frac{u_j}{u_i}, \quad j \in \{1, \dots, i-1, i+1, \dots, d\}. \tag{15}$$

In other words, we are able to obtain a fair linear model without any other constraint and by using a representation that has one feature fewer than the original one[3].

## 4 Experiments

In this section, we present numerical experiments with the proposed method on one synthetic and five real datasets. The aim of the experiments is threefold. First, we show that our approach is effective in selecting a fair model, incurring only a moderate loss in accuracy. Second, we provide an empirical study of the properties of the method, which supports our theoretical observations in Section 2. Third, we highlight the generality of our approach by showing that it can be used effectively within other linear models such as Lasso SVM for classification.

We use our approach with $\epsilon{=}0$ in order to simplify the hyperparameter selection procedure. For the sake of completeness, a set of results for different values of $\epsilon$ is presented in the supplementary material and briefly we comment on these below. In all the experiments, we collect statistics concerning the classification accuracy and DEO of the selected model. We recall that the DEO is defined in Eq. (8) and is the absolute difference of the true positive rate of the classifier applied to the two groups. In all experiments, we performed a 10-fold cross validation (CV) to select the best hyperparameters[4]. For the Arrhythmia, COMPAS, German and Drug datasets, this procedure is repeated 10 times, and we reported the average performance on the test set alongside its standard deviation. For the Adult dataset, we used the provided split of train and test sets. Unless otherwise stated, we employ two steps in the 10-fold CV procedure. In the first step, the value of the hyperparameters with highest accuracy is identified. In the second step, we shortlist all the hyperparameters with accuracy close to the best one (in our case, above $90\%$ of the best accuracy). Finally, from this list, we select the hyperparameters with the lowest DEO. This novel validation procedure, that we wil call NVP, is a sanity-check to ensure that fairness cannot be achieved by a simple modification of hyperparameter selection procedure. The code of our method is available at: `https://github.com/jmikko/fair_ERM`.

**Synthetic Experiment.** The aim of this experiment is to study the behavior of our method, in terms of both DEO and classification accuracy, in comparison to standard SVM (with our novel validation procedure). To this end, we generated a synthetic binary classification dataset with two sensitive groups in the following manner. For each group in the class $-1$ and for the group $a$ in the class $+1$, we generated 1000 examples for training and the same amount for testing. For the group $b$ in the class $+1$, we generated 200 examples for training and the same number for testing. Each set of examples is sampled from a 2-dimensional isotropic Gaussian distribution with different mean $\mu$ and variance $\sigma^2$: (i) Group $a$, Label $+1$: $\mu = (-1, -1)$, $\sigma^2 = 0.8$; (ii) Group $a$, Label $-1$: $\mu = (1, 1)$, $\sigma^2 = 0.8$; (iii) Group $b$, Label $+1$: $\mu = (0.5, -0.5)$, $\sigma^2 = 0.5$; (iv) Group $b$, Label $-1$: $\mu = (0.5, 0.5)$, $\sigma^2 = 0.5$. When a standard machine learning method is applied to this toy dataset, the generated model is unfair with respect to the group $b$, in that the classifier tends to negatively classify the examples in this group.

We trained different models, varying the value of the hyperparameter $C$, and using the standard linear SVM and our linear method. Figure 1 (Left) shows the performance of the various generated models with respect to the classification error and DEO on the test set. Note that our method generated models that have an higher level of fairness, maintaining a good level of accuracy. The grid in the plots emphasizes the fact that both the error and DEO have to be simultaneously considered in the evaluation of a method. Figure 1 (Center and Left) depicts the histogram of the values of $\langle \boldsymbol{w}, \boldsymbol{x} \rangle$ (where $\boldsymbol{w}$ is the generated model) for test examples with true label equal to $+1$ for each of the two groups. The results are reported both for our method (Right) and standard SVM (Center). Note that our method generates a model with a similar true positive rate among the two groups (i.e. the areas of the value when the horizontal axis is greater than zero are similar for groups $a$ and $b$). Moreover,

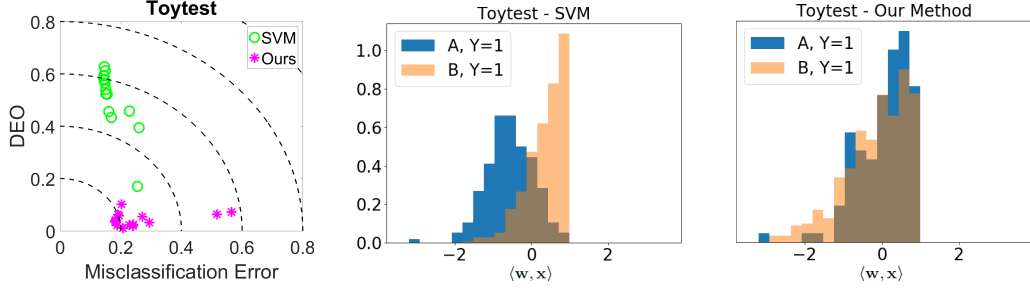

Figure 1: Left: Test classification error and DEO for different values of hyperparameter $C$ for standard linear SVM (green circles) and our modified linear SVM (magenta stars). Center and Rights: Histograms of the distribution of the values $\langle \boldsymbol{w}, \boldsymbol{x} \rangle$ for the two groups ($a$ in blue and $b$ in light orange) for test examples with label equals to $+1$. The results are collected by using the optimal validated model for the classical linear SVM (Center) and for our linear method (Right).

| Method | Arrhythmia | | COMPAS | | Adult | | German | | Drug | |
|---|---|---|---|---|---|---|---|---|---|---|
| | ACC | DEO | ACC | DEO | ACC | DEO | ACC | DEO | ACC | DEO |
| *s not included in $\boldsymbol{x}$* | | | | | | | | | | |
| Naïve Lin. SVM | 0.75±0.04 | 0.11±0.03 | 0.73±0.01 | 0.13±0.02 | 0.78 | 0.10 | 0.71±0.06 | 0.16±0.04 | 0.79±0.02 | 0.25±0.03 |
| Lin. SVM | 0.71±0.05 | 0.10±0.03 | 0.72±0.01 | 0.12±0.02 | 0.78 | 0.09 | 0.69±0.04 | 0.11±0.10 | 0.79±0.02 | 0.25±0.04 |
| Hardt | - | - | - | - | - | - | - | - | - | - |
| Zafar | 0.67±0.03 | 0.05±0.02 | 0.69±0.01 | 0.10±0.08 | 0.76 | 0.05 | 0.62±0.09 | 0.13±0.10 | 0.66±0.03 | 0.06±0.06 |
| Lin. Ours | 0.75±0.05 | 0.05±0.02 | 0.73±0.01 | 0.07±0.02 | 0.75 | 0.01 | 0.69±0.04 | 0.06±0.03 | 0.79±0.02 | 0.10±0.06 |
| Naïve SVM | 0.75±0.04 | 0.11±0.03 | 0.72±0.01 | 0.14±0.02 | 0.80 | 0.09 | 0.74±0.05 | 0.12±0.05 | 0.81±0.02 | 0.22±0.04 |
| SVM | 0.71±0.05 | 0.10±0.03 | 0.73±0.01 | 0.11±0.02 | 0.79 | 0.08 | 0.74±0.03 | 0.10±0.06 | 0.81±0.02 | 0.22±0.03 |
| Hardt | - | - | - | - | - | - | - | - | - | - |
| Ours | 0.75±0.05 | 0.05±0.02 | 0.72±0.01 | 0.08±0.02 | 0.77 | 0.01 | 0.73±0.04 | 0.05±0.03 | 0.79±0.03 | 0.10±0.05 |
| *s included in $\boldsymbol{x}$* | | | | | | | | | | |
| Naïve Lin. SVM | 0.79±0.06 | 0.14±0.03 | 0.76±0.01 | 0.17±0.02 | 0.81 | 0.14 | 0.71±0.06 | 0.17±0.05 | 0.81±0.02 | 0.44±0.03 |
| Lin. SVM | 0.78±0.07 | 0.13±0.04 | 0.75±0.01 | 0.15±0.02 | 0.80 | 0.13 | 0.69±0.04 | 0.11±0.10 | 0.81±0.02 | 0.41±0.06 |
| Hardt | 0.74±0.06 | 0.07±0.04 | 0.67±0.01 | 0.21±0.09 | 0.80 | 0.10 | 0.61±0.15 | 0.15±0.13 | 0.77±0.02 | 0.22±0.09 |
| Zafar | 0.71±0.03 | 0.03±0.02 | 0.69±0.02 | 0.10±0.06 | 0.78 | 0.05 | 0.62±0.09 | 0.13±0.11 | 0.69±0.03 | 0.02±0.07 |
| Lin. Ours | 0.79±0.07 | 0.04±0.03 | 0.76±0.01 | 0.04±0.03 | 0.77 | 0.01 | 0.69±0.04 | 0.05±0.03 | 0.79±0.02 | 0.05±0.03 |
| Naïve SVM | 0.79±0.06 | 0.14±0.04 | 0.76±0.01 | 0.18±0.02 | 0.84 | 0.18 | 0.74±0.05 | 0.12±0.05 | 0.82±0.02 | 0.45±0.04 |
| SVM | 0.78±0.06 | 0.13±0.04 | 0.73±0.01 | 0.14±0.02 | 0.82 | 0.14 | 0.74±0.03 | 0.10±0.06 | 0.81±0.02 | 0.38±0.03 |
| Hardt | 0.74±0.06 | 0.07±0.04 | 0.71±0.01 | 0.08±0.01 | 0.82 | 0.11 | 0.71±0.03 | 0.11±0.18 | 0.75±0.11 | 0.14±0.08 |
| Ours | 0.79±0.09 | 0.03±0.02 | 0.73±0.01 | 0.05±0.03 | 0.81 | 0.01 | 0.73±0.04 | 0.05±0.03 | 0.80±0.03 | 0.07±0.05 |

Table 1: Results (average ± standard deviation, when a fixed test set is not provided) for all the datasets, concerning accuracy (ACC) and DEO .

due to the simplicity of the toy test, the distribution with respect to the two different groups is also very similar when our model is used.

**Real Data Experiments.** We next compare the performance of our model to set of different methods on 5 publicly available datasets: Arrhythmia, COMPAS, Adult, German, and Drug. A description of the datasets is provided in the supplementary material. These datasets have been selected from the standard databases of datasets (UCI, mldata and Fairness-Measures[5]). We considered only datasets with a DEO higher than 0.1, when the model is generated by an SVM validated with the NVP. For this reason, some of the commonly used datasets have been discarded (e.g. Diabetes, Heart, SAT, PSU-Chile, and SOEP). We compared our method both in the linear and not linear case against: (i) Naïve SVM, validated with a standard nested 10-fold CV procedure. This method ignores fairness in the validation procedure, simply trying to optimize accuracy; (ii) SVM with the NVP. As noted above, this baseline is the simplest way to inject the fairness into the model; (iii) Hardt method [2] applied to the best SVM; (iv) Zafar method [3], implemented with the code provided by the authors for the linear case[6]. Concerning our method, in the linear case, it exploits the preprocessing presented in Section 3.

Table 1 shows our experimental results for all the datasets and methods both when $s$ is included in $\boldsymbol{x}$ or not. This result suggests that our method performs favorably over the competitors in that it

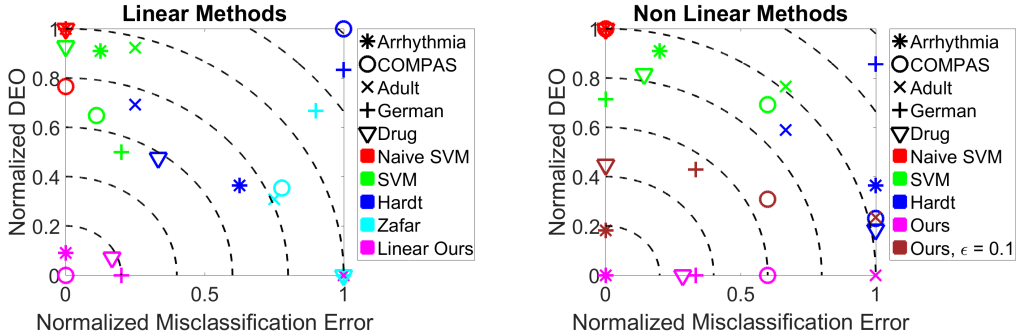

Figure 2: Results of Table 1 of linear (left) and nonlinear (right) methods, when the error and the DEO are normalized in $[0,1]$ column-wise and when $s$ is included in $x$. Different symbols and colors refer to different datasets and method respectively. The closer a point is to the origin, the better the result is.

decreases DEO substantially with only a moderate loss in accuracy. Moreover having $s$ included in $x$ increases the accuracy but - for the methods without the specific purpose of producing fairness models - decreases the fairness. On the other hand, having $s$ included in $x$ ensures to our method the ability of improve the fairness by exploiting the value of $s$ also in the prediction phase. This is to be expected, since knowing the group membership increases our information but also leads to behaviours able to influence the fairness of the predictive model. In order to quantify this effect, we present in Figure 2 the results of Table 1 of linear (left) and nonlinear (right) methods, when the error (one minus accuracy) and the DEO are normalized in $[0,1]$ column-wise both when the $s$ is included and not included in $x$. In the figure, different symbols and colors refer to different datasets and methods, respectively. The closer a point is to the origin, the better the result is. The best accuracy is, in general, reached by using the Naïve SVM (in red) both for the linear and nonlinear case. This behavior is expected due to the absence of any fairness constraint. On the other hand, Naïve SVM has unsatisfactory levels of fairness. Hardt [2] (in blue) and Zafar [3] (in cyan, for the linear case) methods are able to obtain a good level of fairness but the price of this fair model is a strong decrease in accuracy. Our method (in magenta) obtains similar or better results concerning the DEO preserving the performance in accuracy. In particular in the nonlinear case, our method reaches the lowest levels of DEO with respect to all the methods. For the sake of completeness, in the nonlinear (bottom) part of Figure 2, we show our method when the parameter $\epsilon$ is set to $0.1$ (in brown) instead of $0$ (in magenta). As expected, the generated models are less fair with a (small) improvement in the accuracy. An in depth analysis of the role of $\epsilon$ is presented in supplementary material.

**Application to Lasso.** Due to the particular proposed methodology, we are able in principle to apply our method to any learning algorithm. In particular, when the algorithm generates a linear model we can exploit the data preprocessing in Eq. (15), to directly impose fairness in the model. Here, we show how it is possible to obtain a sparse and fair model by exploiting the standard Lasso algorithm in synergy with this preprocessing step. For this purpose, we selected the Arrhythmia dataset as the Lasso works well in a high dimensional / small sample setting. We performed the same experiment described above, where we used the Lasso algorithm in place of the SVM. In this case, by Naïve Lasso, we refer to the Lasso when it is validated with a standard nested 10-fold CV procedure, whereas by Lasso we refer to the standard Lasso with the NVP outlined above. The method of [2] has been applied to the best Lasso model. Moreover, we reported the results obtained using Naïve Linear SVM and Linear SVM. We also repeated the experiment by using a reduced training set in order to highlight the effect of the sparsity. Table 2 reported in the supplementary material shows the results in the case when $s$ is included in $x$. It is possible to note that, reducing the training sets, the generated models become less fair (i.e. the DEO increases). Using our method, we are able to maintain a fair model reaching satisfactory accuracy results.

**The Value of $\hat{\Delta}$.** Finally, we show experimental results to highlight how the hypothesis of Proposition 1 (Section 2.2) are reasonable in the real cases. We know that, if the hypothesis of inequality (11) are satisfied, the linear loss based fairness is close to the EO. Specifically, these two quantities are closer when $\hat{\Delta}$ is small. We evaluated $\hat{\Delta}$ for benchmark and toy datasets. The obtained results

are in Table 3 of supplementary material, where $\hat{\Delta}$ has the order of magnitude of $10^{-2}$ in all the datasets. Consequently, our method is able to obtain a good approximation of the DEO.

Arrhythmia dataset

| Method | Accuracy | DEO | Selected Features |
|---|---|---|---|
| Naïve Lin. SVM | $0.79 \pm 0.06$ | $0.14 \pm 0.03$ | - |
| Linear SVM | $0.78 \pm 0.07$ | $0.13 \pm 0.04$ | - |
| Naïve Lasso | $0.79 \pm 0.07$ | $0.11 \pm 0.04$ | $22.7 \pm 9.1$ |
| Lasso | $0.74 \pm 0.04$ | $0.07 \pm 0.04$ | $5.2 \pm 3.7$ |
| Hardt | $0.71 \pm 0.05$ | $0.04 \pm 0.06$ | $5.2 \pm 3.7$ |
| Our Lasso | $0.77 \pm 0.02$ | $0.03 \pm 0.02$ | $7.5 \pm 2.0$ |

Arrhythmia dataset - Training set reduced by 50%

| Method | Accuracy | DEO | Selected Features |
|---|---|---|---|
| Naïve Lin. SVM | $0.69 \pm 0.03$ | $0.16 \pm 0.03$ | - |
| Linear SVM | $0.68 \pm 0.03$ | $0.15 \pm 0.03$ | - |
| Naïve Lasso | $0.73 \pm 0.04$ | $0.15 \pm 0.06$ | $14.1 \pm 6.6$ |
| Lasso | $0.70 \pm 0.04$ | $0.09 \pm 0.05$ | $7.9 \pm 8.0$ |
| Hardt | $0.67 \pm 0.06$ | $0.08 \pm 0.07$ | $7.9 \pm 8.0$ |
| Our Lasso | $0.71 \pm 0.04$ | $0.03 \pm 0.04$ | $9.0 \pm 7.3$ |

Table 2: Results (average $\pm$ standard deviation) when the model is the Lasso, concerning accuracy, DEO and the number of features with weight bigger than $10^{-8}$ over the 279 original features. The experiment has been repeated also reducing the training set. In this case $s$ is included in $x$.

| Dataset | $\hat{\Delta}$ |
|---|---|
| Toytest | 0.03 |
| Toytest Lasso | 0.02 |
| Arrhythmia | 0.03 |
| COMPAS | 0.04 |
| Adult | 0.06 |
| German | 0.05 |
| Drug | 0.03 |

Table 3: The $\hat{\Delta}$ for the exploited datasets. A smaller $\hat{\Delta}$ means a better approximation of the DEO in our method.

## 5  Conclusion and Future Work

We have presented a generalized notion of fairness, which encompasses previously introduced notion and can be used to constrain ERM, in order to learn fair classifiers. The framework is appealing both theoretically and practically. Our theoretical observations provide a statistical justification for this approach and our algorithmic observations suggest a way to implement it efficiently in the setting of kernel methods. Experimental results suggest that our approach is promising for applications, generating models with improved fairness properties while maintaining classification accuracy. We close by mentioning directions of future research. On the algorithmic side, it would be interesting to study whether our method can be improved by other relaxations of the fairness constraint beyond the linear loss used here. Applications of the fairness constraint to multi-class classification or to regression tasks would also be valuable. On the theory side, it would be interesting to study how the choice of the parameter $\epsilon$ affects the statistical performance of our method and derive optimal accuracy-fairness trade-off as a function of this parameter.

#### Acknowledgments

We wish to thank Amon Elders, Theodoros Evgeniou and Andreas Maurer for useful comments. This work was supported in part by SAP SE and the EPSRC.

## Footnotes

[1]The extension to multiple groups (e.g. ethnic groups) is briefly discussed in the supplementary material, Section I.

[2]In supplementary material we derive the dual of Problem (14) when $\ell_c$ is the Hinge loss.

[3]In supplementary material is reported the generalization of this argument to kernel for SVM.

[4]The regularization parameter $C$ (for both SVM and our method) with 30 values, equally spaced in logarithmic scale between $10^{-4}$ and $10^4$; we used both the linear or RBF kernel (i.e. for two examples $\boldsymbol{x}$ and $\boldsymbol{z}$, the RBF kernel is $e^{-\gamma\|\boldsymbol{x}-\boldsymbol{z}\|^2}$) with $\gamma \in \{0.001, 0.01, 0.1, 1\}$. In our case, $C = \frac{1}{2\lambda}$ of Eq. (14).

[5]Fairness-Measures website: `fairness-measures.org`

[6]Python code for [3]: `https://github.com/mbilalzafar/fair-classification`

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
