[Supplementary Material · PaperSup.pdf]

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

## Supplementary Material

## A  Extension to other Fairness Measures

Several notions of fairness of a classifier have been proposed in the literature. In this work we focus on the notion of Equal Opportunity (EOp). However, our approach naturally generalizes to other notions of fairness which employ constraints based on the expectation or the conditional expectation of the classifier. We discuss here the case of Equalized Odds (EOd) and Demographic Parity (DP). For $\diamond \in \{-, +\}$, the EOp$^\diamond$ constraint is defined as [2]

$$\mathbb{P}\{f(\boldsymbol{x})>0 \mid s=1, y=\diamond 1\}=\cdots=\mathbb{P}\{f(\boldsymbol{x})>0 \mid s=k, y=\diamond 1\},$$

where, for generality, we consider a multiple valued sensitive feature $s \in \mathcal{S} = \{1, \cdots, k\}$. The EOd, instead is just the concurrent verification of the EOp$^+$ and EOp$^-$, then

$$\mathbb{P}\{f(\boldsymbol{x})>0 \mid s=1, y=\diamond 1\}=\cdots=\mathbb{P}\{f(\boldsymbol{x})>0 \mid s=k, y=\diamond 1\}, \quad \forall \diamond \in \{-, +\}.$$

For what concern the DP, we have to satisfy the constraint

$$\mathbb{P}\{f(\boldsymbol{x})>0 \mid s=1\}=\cdots=\mathbb{P}\{f(\boldsymbol{x})>0 \mid s=k\}.$$

Since a model $f$, in general, will not be able to exactly fulfill the EOp$^\diamond$ with $\diamond \in \{-, +\}$ nor the EOd constraints we define the Difference of EOp$^\diamond$ (DEOp$^\diamond$) with $\diamond \in \{-, +\}$ as

$$\text{DEOp}^\diamond = \sum_{t \in \mathcal{S}} \left| \mathbb{P}\{\hat{y} = y | s=t, y=\diamond 1\} - \frac{1}{|\mathcal{S}|} \sum_{t' \in \mathcal{S}} \mathbb{P}\{\hat{y} = y | s=t', y=\diamond 1\} \right|, \quad (16)$$

where $\hat{y} = \text{sign}(f(\boldsymbol{x}, s))$. Consequently, the Difference of EOd (DEOd) is defined as

$$\text{DEOd} = \frac{\text{DEOp}^+ + \text{DEOp}^-}{2}. \quad (17)$$

Finally, the Difference of DP (DDP) is defined as

$$\text{DDP} = \sum_{t \in \mathcal{S}} \left| \mathbb{P}\{\hat{y} = y | s=t\} - \frac{1}{|\mathcal{S}|} \sum_{t' \in \mathcal{S}} \mathbb{P}\{\hat{y} = y | s=t'\} \right|. \quad (18)$$

Note that it should be apparent that the above fairness measures can be reformulated in terms of differences between risks and then using the linear loss the constraints become convex. The generalization properties of the resulting FERM method can be derived following the same reasoning in Section 2.2.

## B  Proofs

**Proof of Theorem 1.** We first use Eq. (5) to conclude that, with probability at least $1 - 2\delta$,

$$\sup_{f \in \mathcal{F}} \left| \left| L^{+,a}(f) - L^{+,b}(f) \right| - \left| \hat{L}^{+,a}(f) - \hat{L}^{+,b}(f) \right| \right| \leq \sum_{g \in \{a,b\}} B(\delta, n^{+,g}, \mathcal{F}). \quad (19)$$

This inequality in turn implies that, with probability at least $1 - 2\delta$, it holds that

$$\left\{ f : f \in \mathcal{F}, \left| L^{+,a}(f) - L^{+,b}(f) \right| \leq \epsilon \right\} \subseteq \left\{ f : f \in \mathcal{F}, \left| \hat{L}^{+,a}(f) - \hat{L}^{+,b}(f) \right| \leq \hat{\epsilon} \right\}. \quad (20)$$

Now, in order to prove the first statement of the theorem, let us decompose the excess risk as

$$L(\hat{f}) - L(f^*) = L(\hat{f}) - \hat{L}(\hat{f}) + \hat{L}(\hat{f}) - \hat{L}(f^*) + \hat{L}(f^*) - L(f^*).$$

Inequality (20) implies that $\hat{L}(\hat{f}) - \hat{L}(f^*) \leq 0$ with probability at least $1 - 2\delta$ and consequently with probability at least $1 - 2\delta$ it holds that

$$L(\hat{f}) - L(f^*) \leq L(\hat{f}) - \hat{L}(\hat{f}) + \hat{L}(f^*) - L(f^*).$$

The first statement now follows by Eq. (5). As for the second statement, its proof consists in exploiting the results of Eqns. (19) and (20) together with a union bound. ∎

**Proof of Proposition 1.** The proof of the first statement follows directly by the inequality $\ell_h(f(\boldsymbol{x}), y) \leq \ell_c(f(\boldsymbol{x}), y)$. In order to prove the second statement, we first note that

$$\left| \hat{L}_l^{+,a}(f) - \hat{L}_l^{+,b}(f) \right| = \frac{1}{2} \left| \hat{\mathbb{E}}\left[ f(\boldsymbol{x}) | y = 1, s = a \right] - \hat{\mathbb{E}}\left[ f(\boldsymbol{x}) | y = 1, s = b \right] \right|.$$

By applying the same reasoning to $\left| \hat{L}_h^{+,a}(f) - \hat{L}_h^{+,b}(f) \right|$ and by exploiting inequality (11) the result follows. ∎

## C  Literature Review of Fairness Methods

In this section, we provide a brief analysis of the different existing methods concerning fairness. We show our findings in Table 4, where the rows represent properties, characteristics and experimental results of different fairness methods. The columns represent the different algorithms and, specifically, the first column is our approach. We think that, at this stage of development of fairness in machine learning, a clear understanding of the differences and similarities among the current available algorithms is a fundamental step. Table 4 describes, in the first row, the family of the different methods, following the taxonomy defined in this paper (see Section 1). The following 8 rows describe general properties of the methods, as for example the convexity of the approach, the convergence of the learning phase or the consistency with respect to the risk and the fairness notion. The next 9 rows describes the presence of a specific comparison between methods and, finally, in the last row the availability of the code online is analyzed.

| Ref. | Ours | [27] | [7] | [18, 19] | [20] | [3] | [28] | [32] | [4] | [21] | [1] | [22] | [15] | [16] | [23] | [2] | [24] | [33] | [25] | [17] | [29] | [30] | [34] | [38] |
|---|---|---|---|---|---|---|---|---|---|---|---|---|---|---|---|---|---|---|---|---|---|---|---|---|
| Method Family | 2&3 | 3 | 3 | 2 | 2 | 2 | 3 | 2 | 3 | 2 | 2 | 2 | 1 | - | 2 | 1 | 2 | 2 | 2 | 1 | 3 | 3 | 2 | 2 |
| Classification | x | x | x | x | x | x | x |  | x | x | x | x | x | x | x | x | x |  | x | x | x | x | x | x |
| New Fairness Notions | x | x | x | x | x | x | x | x | x | x | x | x | x |  | x | x | x | x | x | x | x | x | x |  |
| Use of EO | x |  |  | x |  |  |  |  | x | x | x |  |  |  | x |  |  |  |  |  |  |  |  | x |
| Convex Approach | x | x |  | x* |  |  | x |  |  | x |  |  |  |  | x |  | x | x |  |  | x |  | x |  |
| Convergence Learning | x | x | x | x |  |  |  | x |  |  |  | x |  |  |  |  |  |  | x |  |  |  | x |  |
| Consistency Risk-Fairness | x |  |  | x | x |  |  |  |  | x | x |  |  |  |  |  |  |  |  |  |  |  |  |  |
| Experimental Results | x | x | x | x |  | x | x | x | x | x | x | x | x | x | x | x | x | x | x | x | x | x |  | x |
| Epsilon validate | x |  |  |  |  |  |  |  |  |  |  |  |  |  |  |  |  |  |  |  |  |  |  |  |
| Exp. w.r.t. [2] | x |  |  | x |  | x |  |  |  |  |  | x |  |  | x |  |  |  |  |  |  |  |  | x |
| Exp. w.r.t. [3] | x |  |  |  |  |  |  |  |  |  |  | x |  |  | x |  |  |  |  |  |  |  |  | x |
| Exp. w.r.t. [29] |  |  |  |  | x |  |  |  |  |  |  |  |  |  |  |  |  |  |  |  |  |  |  |  |
| Exp. w.r.t. Baseline in [3] |  |  |  |  | x |  |  |  |  |  |  |  |  |  | x |  |  |  |  |  |  |  |  |  |
| Exp. w.r.t. [28] |  |  |  |  |  |  |  |  | x |  |  |  |  |  |  |  |  |  |  | x |  | x |  |  |
| Exp. w.r.t. [25] |  |  |  |  |  |  |  |  | x |  |  |  |  |  |  |  | x |  | x |  |  |  |  |  |
| Exp. w.r.t. [30] |  |  |  |  |  |  |  |  |  |  |  |  |  |  |  |  | x |  |  |  |  |  |  |  |
| Exp. w.r.t. [4] |  |  |  |  |  |  |  |  |  |  |  |  |  |  |  |  |  |  |  | x |  |  |  |  |
| Code Available | x |  | x |  |  | x |  | x |  |  |  | x |  |  | x |  | x |  |  | x |  |  |  |  |

Table 4: A summary of the characteristics of the different methods concerning fairness. The symbol 'x' means the presence of a property (row) for a specific method (column). x*: the theoretical results however do not correspond to their convex method.

## D  Datasets

In the following the datasets used in Section 4 are presented, outlining their tasks, type of features and source of data. Table 5 provide a summary of the datasets statistics.

- *Arrhythmia*: from UCI repository, this database contains 279 attributes concerning the study of H. Altay Guvenir. The aim is to distinguish between the presence and absence of cardiac arrhythmia and to classify it in one of the 16 groups. In our case, we changed the task with the binary classification between "Class 01" (i.e. "Normal") against the other 15 classes (different classes of arrhythmia).

- *COMPAS* (Correctional Offender Management Profiling for Alternative Sanctions): it is a popular commercial algorithm used by judges and parole officers for scoring criminal defendant's likelihood of reoffending (recidivism). It has been shown that the algorithm is biased in favor of white defendants based on a 2 year follow up study. This dataset contains variables used by the COMPAS algorithm in scoring defendants, along with their outcomes within 2 years of the decision, for over 10000 criminal defendants in Broward County, Florida. In the original data, 3 subsets are provided. We concentrate on the one that includes only violent recidivism[7].

- *Adult*: from UCI repository, this database contains 14 features concerning demographic characteristics of 45222 instances (32561 for training and 12661 for test). The task is to predict if a person has an income per year that is more (or less) than 50000 $. Concerning the Adult dataset we used the provided training and test sets.

| Dataset | Examples | Features | Sensitive Variable |
|---|---|---|---|
| Arrhythmia | 452 | 279 | Gender |
| COMPAS | 6172 | 10 | Ethnicity |
| Adult | 32561, 12661 | 12 | Gender |
| German | 1700 | 20 | Foreign |
| Drug | 1885 | 11 | Ethnicity |

Table 5: Datasets statistics (for Adult we reported the number of train testing examples provided) and their sensitive feature. Gender considers the two groups as male and female; ethnicity considers the ethnic groups white and other ethnic groups; foreign considers being or not being a foreign person.

Figure 3: Results concerning the Drug dataset for Naïve SVM, Hard method and our method with different values of $\epsilon$.

- *German*: it is a dataset where the task is to classify people described by a set of 20 features (7 numerical, 13 categorical) as good or bad credit risks. The features are related to the economical situation of the person, as for example: credit history and amount, saving account and bonds, year of the present employment, property and others. Moreover, a set of features is concerning personal information, e.g. age, gender, if the person is a foreign, and personal status.

- *Drug*: this dataset contains records for 1885 respondents. Each respondent is described by 12 features: Personality measurements which include NEO-FFI-R (neuroticism, extraversion, openness to experience, agreeableness, and conscientiousness), BIS-11 (impulsivity), and ImpSS (sensation seeking), level of education, age, gender, country of residence and ethnicity. All input attributes are originally categorical and are quantified. After quantification values of all input features can be considered as real-valued. In addition, participants were questioned concerning their use of 17 legal and illegal drugs and one fictitious drug (Semeron) which was introduced to identify over-claimers. For each drug, the respondents have to select one of the answers: never used the drug, used it over a decade ago, or in the last decade, year, month, week, or day. In this sense, this dataset contains 18 classification problems, each one with seven classes: "Never Used", "Used over a Decade Ago", "Used in Last Decade", "Used in Last Year", "Used in Last Month", "Used in Last Week", and "Used in Last Day". We make the problem number 16 (concerning heroin) a binary problem by exploiting the task "Never used" versus "Others" (i.e. "Used").

# E  Varying the Value of $\epsilon$

In this section we present a set of experiments, as a proof of concept, that our selection of $\epsilon = 0$ for our method is reasonable and study the impact of different values of $\epsilon$ have concerning DEO and accuracy performance.

We follow the same experimental setting presented in Section 4 for the Drug dataset, implementing our nonlinear method with $\epsilon$ equals to $0, 0.01, 0.1, 0.2, 0.3$. The results of this experiment are presented in Figure 3, where we show also the results for Naïve SVM and Hard method. It is possible

Figure 4: Results of Table 1 of linear (left) and nonlinear (right) methods, when the error and the DEO are normalized in $[0, 1]$ column-wise and when $s$ is not included in $\boldsymbol{x}$. Different symbols and colors refer to different datasets and method respectively. The closer a point is to the origin, the better the result is.

Figure 5: Error (Left), DEO (Middle) and our approximation of the DEO (Right) for the German dataset by using our method. For the three images, the x-axis is the logarithm of the regularization hyperparameter $C$ and the y-axis is the logarithm of the RBF hyperparameter $\gamma$.

to note how increasing the value of $\epsilon$, our model has smaller error but stronger unfairness (i.e. higher DEO).

## F  Visualization of the Results of Table 1

In Figure 4 we reported the equivalent of Figure 2 for the case when $s$ is not included in $\boldsymbol{x}$. Note that we can reach the same conclusions drown for Table 1 and Figure 2.

## G  Approximation of the DEO

In this section, we numerically show the difference between the DEO and our approximation of it. Figure 5 compares the DEO with our approximation of the DEO and the classification error. We collected these results for the German dataset on the validation set, changing the two hyperparameters $C$ and $\gamma$ (in the nonlinear case). We can note how our approximation of the DEO is empirically similar to the original DEO. It is interesting to highlight that, a correct approximation of the DEO is particularly important where the error is low.

## H  Dual Problem for SVM with Fairness Constraint

We follow the usual approach to derive the dual problem for SVMs, which uses the method of Lagrange multipliers [43]. We define the Lagrangian function

$$\mathcal{L}(\boldsymbol{w}, \boldsymbol{\xi}, \boldsymbol{\alpha}, \boldsymbol{\beta}, \boldsymbol{\rho}) = \frac{1}{2} \langle \boldsymbol{w}, \boldsymbol{w} \rangle + C \sum_{i=1}^{n} \xi_i - \sum_{i=1}^{n} \alpha_i (y_i \langle \boldsymbol{\phi}(\boldsymbol{x}_i), \boldsymbol{w} \rangle - 1 + \xi_i) - \beta_i \xi_i +$$
$$\rho_1(\langle \boldsymbol{w}, \boldsymbol{u} \rangle - \epsilon) - \rho_2(\langle \boldsymbol{w}, \boldsymbol{u} \rangle + \epsilon) \quad (21)$$

where $\boldsymbol{\alpha}, \boldsymbol{\beta}$ and $\boldsymbol{\rho}$ are the vector of Lagrange multipliers and are constrained to be nonnegative. We set the derivative of the Lagrangian with respect to the primal variables $\boldsymbol{w}$ and $\boldsymbol{\xi}$ equal to zero. In the latter case we obtain that

$$C - \alpha_i - \beta_i = 0 \tag{22}$$

from which we can remove the variable $\beta_i$ in place of the constraint $\alpha_i \leq C$. In the former case we obtain the expression for $\boldsymbol{w}$,

$$\boldsymbol{w} = \sum_{i=1}^{n} \alpha_i y_i \boldsymbol{\phi}(\boldsymbol{x}_i) + (\rho_1 - \rho_2)\boldsymbol{u}. \tag{23}$$

Using (22) and (23) in (21) we obtain the expression

$$-\frac{1}{2}\left\| \sum_{i=1}^{n} \alpha_i y_i \boldsymbol{\phi}(\boldsymbol{x}_i) + (\rho_1 - \rho_2)\boldsymbol{u} \right\|^2 + \sum_{i=1}^{n} \alpha_i - \epsilon(\rho_1 + \rho_2). \tag{24}$$

The dual problem is then to maximize this quantity subject to the constraints that $\alpha_i \in [0, C]$ and $\rho_1, \rho_2 \geq 0$.

The KKT conditions are

$$\alpha_i(1 - y_i\langle \boldsymbol{w}, \boldsymbol{\phi}(\boldsymbol{x}_i)\rangle - 1 + \xi_i) = 0 \tag{25}$$
$$C - \alpha_i \xi_i = 0 \tag{26}$$
$$\rho_1(\langle \boldsymbol{w}, \boldsymbol{u}\rangle - \epsilon) = 0 \tag{27}$$
$$\rho_2(\langle \boldsymbol{w}, \boldsymbol{u}\rangle + \epsilon) = 0. \tag{28}$$

Clearly at most one of the variables $\rho_1$ and $\rho_2$ can be strictly positive. We may then let $\rho = \rho_1 - \rho_2$ and rewrite the objective function as

$$-\frac{1}{2}\left\| \sum_{i=1}^{n} \alpha_i y_i \boldsymbol{\phi}(\boldsymbol{x}_i) + \rho\boldsymbol{u} \right\|^2 + \sum_{i=1}^{n} \alpha_i - \epsilon|\rho| \tag{29}$$

and optimize over $\boldsymbol{\alpha} \in [0, C]^n$ and $\rho \in \mathbb{R}$. It is interesting to study this problem when $\epsilon = 0$. In this case we can easily solve for $\rho$ obtaining the simplified objective

$$-\frac{1}{2}\left\| \sum_{i=1}^{n} \alpha_i y_i (I - P)\boldsymbol{\phi}(\boldsymbol{x}_i)) \right\|^2 + \sum_{i=1}^{n} \alpha_i$$

where $P$ is the orthogonal projection along the direction of $\boldsymbol{u}$, that is $P = \frac{\boldsymbol{u}}{\|\boldsymbol{u}\|} \otimes \frac{\boldsymbol{u}}{\|\boldsymbol{u}\|}$. This is equivalent to use the standard SVM with the kernel

$$\widetilde{\kappa}(\boldsymbol{x}, \boldsymbol{t}) = \langle \boldsymbol{\phi}(\boldsymbol{x}), (I - P)\boldsymbol{\phi}(\boldsymbol{t})\rangle = \kappa(\boldsymbol{x}, \boldsymbol{t}) - \frac{\langle \boldsymbol{x}, \boldsymbol{u}\rangle\langle \boldsymbol{t}, \boldsymbol{u}\rangle}{\|\boldsymbol{u}\|^2}$$

In particular if $\boldsymbol{u} = \dfrac{1}{n_a}\displaystyle\sum_{i \in \mathcal{I}^{+,a}} \boldsymbol{x}_i - \dfrac{1}{n_b}\displaystyle\sum_{i \in \mathcal{I}^{+,b}} \boldsymbol{x}_i$, we obtain

$$\widetilde{\kappa}(\boldsymbol{x}, \boldsymbol{t}) = \kappa(\boldsymbol{x}, \boldsymbol{t}) - \frac{\frac{1}{n_a}\sum_{i \in \mathcal{I}^{+,a}} \kappa(\boldsymbol{x}, \boldsymbol{x}_i) - \frac{1}{n_b}\sum_{i \in \mathcal{I}^{+,b}} \kappa(\boldsymbol{x}, \boldsymbol{x}_i)}{\frac{1}{n_a^2}\sum_{i,j \in \mathcal{I}^{+,a}} \kappa(\boldsymbol{x}_i, \boldsymbol{x}_j) + \frac{1}{n_b^2}\sum_{i,j \in \mathcal{I}^{+,b}} \kappa(\boldsymbol{x}_i, \boldsymbol{x}_j) - \frac{2}{n_a n_b}\sum_{i \in \mathcal{I}^{+,a}} \sum_{j \in \mathcal{I}^{+,b}} \kappa(\boldsymbol{x}_i, \boldsymbol{x}_j)}.$$

This new kernel can then be interpreted as a change of feature mapping $\boldsymbol{x} \mapsto (I - P)\boldsymbol{\phi}(\boldsymbol{x}) = \boldsymbol{\phi}(\boldsymbol{x}) - \langle \boldsymbol{\phi}(\boldsymbol{x}), \frac{\boldsymbol{u}}{\|\boldsymbol{u}\|}\rangle \frac{\boldsymbol{u}}{\|\boldsymbol{u}\|}$.

As a final remark, we note that for other proper convex loss functions (e.g. square loss or logistic loss) the dual problem can be derived via Fenchel duality [see e.g. 44]. We leave the full details to a future occasion.

# I Multiple Valued Sensitive Features

Our method presented in Section 3 can be naturally extended to the case that the sensitive variable takes multiple categorical values, that is $s \in \{g_1, \ldots, g_k\}$ for some $k \geq 2$. In particular, when $\epsilon = 0$, the fairness constraint in Problem (4) requires that

$$\hat{L}^{+,g_1}(f) = \hat{L}^{+,g_2}(f) = \cdots = \hat{L}^{+,g_k}(f). \tag{30}$$

Furthermore if the linear loss function is used, these constraints becomes

$$\langle \mathbf{w}, \mathbf{u}_1 - \mathbf{u}_g \rangle = 0, \quad \forall g \in \{g_2, \cdots, g_k\}$$

where we defined, for $g \in \{g_1, \ldots, g_k\}$

$$\mathbf{u}_g = \frac{1}{n^{+,g}} \sum_{i \in \mathcal{I}^{+,g}} \phi(\boldsymbol{x}_i),$$

with $\mathcal{I}^{+,g} = \{i : y_i = 1, s = g\}$ and $n^{+,g} = |\mathcal{I}^{+,g}|$. Thus, we need to satisfy $k - 1$ orthogonality constraints which try to enforce a balance between the different sensitive groups as measured by the barycenters of the within groups positive labeled points. Similar considerations apply when dealing with multiple sensitive features.