[Reviews · NeurIPS 2018]

Reviewer 1



UPDATE: Thanks to the authors for their rebuttal. After reading that and the comments from my colleagues, I am still convinced about the novelty and importance of this work, and choose to keep my score as it is. The authors propose an empirical risk minimization scheme that incorporates fairness constraints. A particular form of fairness constraint, equal opportunity is used, whose relaxation becomes a conditional risk with respect to the sensitive variable. By mandating these conditional risks for various sensitive attributes to be close, fairness is achieved. The authors also show the following: 1. When the relaxation of difference in equal opportunity (relaxed DEO) is set to 0, the fairness requirement becomes an orthogonality constraint. 2. In addition, when f(x) is linear, fairness can be achieved using a simple pre-processing step on the data. 3. Consistency bounds on estimating f from sample data. 4. Extensions to RKHS. The paper is well-written and easy to understand. It has a good mix of theory and practical applications to datasets. I believe such an ERM formulation is a valuable contribution. Here are my questions/comments: 1. In line 101, the authors claim that the relaxation from conditional probability to conditional risk proposed for the EO fairness measure can be easily extended to others. They give equal odds as an example, and surely this extension is conceivable. However, such extensions are possible only for fairness measures that are based on conditional probabilities, which can be noted. 2. In line 190, is the choice of |u_i| arbitrary? Any other |u_i|would work, right? 3. In lines 191 and (15), should the term multiplying x_i be u_j/u_i? 4. Instead of saying “s is inside/not inside x”, you could say “s is included/not included in x”. 5. When applying this in LASSO, how was the accuracy computed, since LASSO will provide a continuous valued output? In LASSO, was s included in x? 6. Typo: “Misclassification error” in the figures. Supplementary: 1. In table 2, [7] has code available. 2. It may be better if the authors provide more details on lines 400 and 401 in supplementary, for the benefit of readers in the fairness community.

Reviewer 2



UPDATE: After reading all reviews and rebuttal, I am more confident in my initial rating. I think, this is a good work and would recommend its acceptance. I also like Table 2 in the supplementary on summary of the characteristics of the different methods concerning fairness. In the missing reference "Recycling privileged learning and distribution matching for fairness", the authors have already used the notion of epsilon fairness (7b), albeit it was not formally defined as in this work. This paper proposes fair classifier learning based on empirical risk minimization framework with fairness constrains (FERM). The authors define the concept of epsilon-fairness, i.e. a fair model commits approximately the same error across the sensitive group on the positive class. This definition accommodates the definition of equality of opportunity (when epsilon=0 and the hard loss function is used to compute the error rate) and can be extended to equalized odds. The authors derive both risk and fairness bounds to show that the FERM method produces statistically consistent estimator (Theorem 1). Furthermore, because the fairness constraints are non-convex, the authors derive convex relaxations of the constraints using the linear loss (1-yf(x))/2, and also use the Hinge loss instead of the hard loss in the risk functional to derive a convex FERM problem (10). The authors also explored the quality of such approximation in terms of classification accuracy and fairness (Proposition 1). The theoretical analysis and the prosed method are appealing. It also has a nice interpretation. When FERM is embedded in the RKHS and eps=0, the learned parameter vector w has to be orthogonal to the difference between the sensitive groups (more precisely, orthogonal to the difference of their barycenters). Because the difference in the sensitive groups is due to sensitive information in the first place, this can be also reinterpreted as w has to be orthogonal to the sensitive information. In the special case with linear kernels, the orthogonality (fairness) constraint can be enforced via simple data pre-processing step (15). The empirical evaluations are convincing. I like that the authors evaluate both scenarios when s is present/not present in x. My only concern with this work is that the main idea of matching the error rates as per definition 1 has been addressed in the last year NIPS paper "Recycling privileged learning and distribution matching for fairness" by Quadrianto and Sharmanska, where the authors matched the distributions over error rates across sensitive groups using MMD criterion. MMD criterion matches potentially infinitely many moments when embedded in the RKHS. To the best of my understanding, using linear loss in fairness constraints, as it is proposed in this work, means matching first order moments (barycenters) in the context of distribution matching. I wonder whether the authors could comment on the relation to this work.

Reviewer 3



UPDATE: "We are aware of these papers and we feel they are less complete works with respect to ours": I do not think I have asked the authors to review these works. Their degree of completeness has been judged already by the venues they have been published at. What I am talking about is a comparison to the very related work, and the fact that you think your paper is superior to these does not, in my opinion, answer the question to the missing comparisons. - The impact of the paper is quite limited. Also, novelty is well beyond the NIPS level. - The linear loss is important, but it rather limits the impact. Most of the practical fairness problems in the literature are not based on linear classifier solutions. - An important and related missing reference is: "Inherent trade-offs in the fair determination of risk scores". - L53: "[2, 15–17] are examples of these methods but 54 no consistency property nor comparison with state-of-the-art proposal are provided." Some of these papers, especially 'On fairness and calibration', are providing rigorous assessment and are being very verbose and convincing in their analysis (and that includes the theoretical one) and on the impossibility of fully satisfying accuracy, fairness and calibration constraints without. My opinion is that this (and it is not the only place in the paper with this theme) claiming that the others have not done the aforementioned issues before is a bit unprofounded. - Just to put them together, the aforementioned paper "Inherent trade-offs in the fair determination of risk scores" along with ref. [15], "On fairness and calibration, NIPS 2017", are suggestions for future joint investigation and its reflection on the current version of the paper. - Writing of the paper and the flow of the ideas therein have a room for improvement. There are also some typos, e.g.: L18: "which does not unfair use additional sensitive information" - Beginning a part of a statement with this reference style without a word [2, 15-17] (L53) is not correct.

Reviewer 4



Overview: This paper presents an approach to fairness in statistical learning based on surrogate losses. They work in a generalization of the "equality of opportunity" (EO) framework of Hardt, Price, and Srebro in which the learner performs an ERM problem subject to a constraint that (perhaps only approximately) enforces equality of opportunity. The EO framework is relaxed so that we require equality of some abstract loss function across protected attributes rather than equality of probability that the classifier outputs a specific class. The EO constraint is also relaxed so that it only needs to hold approximately. After introducing this framework, the authors show relax zero-one losses appearing in the objective or the EO constraint to surrogate losses. Specifically, they show how to bound approximation error when the zero-one loss in the ERM objective is replaced with the hinge loss and the zero-one loss in the EO constraint is replaced with the linear loss. For the special case where the function class is an RKHS they show to directly solve the fair ERM problem under this relaxation. Finally, the authors conduct extensive experiments against comparable fair learning algorithms and find that their surrogate loss approach is competitive for linear classes. Originality and Significance: On the mathematical side, I was somewhat disappointed with the limited scope of the paper's theory results relative to the breadth suggested by the title. In particular, the results appearing in the main body, theorem 1 and proposition 2, use mostly standard calculations from learning theory and in particular surrogate losses. It is indeed unfortunate that proposition 2 is limited to the special case of hinge loss and linear loss, and does not give a general framework. My conclusion here is that in terms of analysis tools there are not many new results or tricks that a broader audience might hope to take away. I am willing to accept however that this paper makes a nice conceptual contribution, namely for the special case of their general problem (hinge/linear loss, linear function class) that is the focus of page 5 onward, the experimental evaluation the authors perform suggests that this method indeed outperforms that of comparable algorithms from Hardt et al. and Zafar et al. So I believe the method has practical merit and could be a useful baseline/benchmark for future development in this area. There is one caveat which is that all the algorithms compared against use the same linear representation. It would not suprise me if using one of the algorithms from [18-26] that direct solves the nonconvex problem for a more expressive function class (eg decision forests) can obtain higher accuracy and EO simultaneously, so this seems like an important benchmark. To make the theory results stronger in the future it would be nice to give comparison results in the vein of proposition 2 for more general loss functions, as well as to try to establish proper classification calibration results in the vein of Bartlett, Jordan, and McAullife. Quality and Clarity: I found the paper easy to read and follow overall. A couple minor notes on the writing: * People not already familiar with surrogate losses may find the discussion on page 4 unclear. * Try to define all the named losses (\ell_h, \ell_c, \ell_l) in the same place to avoid making people search for their definitions. * Please try to find a way to highlight the conclusions of Table 1 visually, eg highlight highest accuracy and EO for each dataset. I appreciate the fairly extensive discussion of the experimental setup in the main paper and appendix.